https://doi.org/10.1038/s41467-019-08984-7　　**OPEN**

# Rapid mantle flow with power-law creep explains deformation after the 2011 Tohoku mega-quake

Ryoichiro Agata [1], Sylvain D. Barbot[2], Kohei Fujita[3,4], Mamoru Hyodo[1], Takeshi Iinuma [1], Ryoko Nakata[1], Tsuyoshi Ichimura[3,4,5] & Takane Hori[1]

The deformation transient following large subduction zone earthquakes is thought to originate from the interaction of viscoelastic flow in the asthenospheric mantle and slip on the megathrust that are both accelerated by the sudden coseismic stress change. Here, we show that combining insight from laboratory solid-state creep and friction experiments can successfully explain the spatial distribution of surface deformation in the first few years after the 2011 $M_\mathrm{w}$ 9.0 Tohoku-Oki earthquake. The transient reduction of effective viscosity resulting from dislocation creep in the asthenosphere explains the peculiar retrograde displacement revealed by seafloor geodesy, while the slip acceleration on the megathrust accounts for surface displacements on land and offshore outside the rupture area. Our results suggest that a rapid mantle flow takes place in the asthenosphere with temporarily decreased viscosity in response to large coseismic stress, presumably due to the activation of power-law creep during the post-earthquake period.

[1] R&D Center for Earthquake and Tsunami, Japan Agency for Marine-Earth Science and Technology, 3173-25, Showa-machi, Kanazawa-ku, Yokohama, Kanagawa 2360001, Japan. [2] Department of Earth Sciences, University of Southern California, 3651 Trousdale Pkwy, Los Angeles, CA 90089-0740, USA. [3] Earthquake Research Institute & Department of Civil Engineering, The University of Tokyo, 1-1-1, Yayoi, Bunkyo-ku, Tokyo 1130032, Japan. [4] Center for Computational Science, RIKEN, 7-1-26, Minatojimaminami-machi, Chuo-ku, Kobe, Hyogo 6500047, Japan. [5] Center for Advanced Intelligence Project, RIKEN, Nihonbashi 1-chome Mitsui Building, 15th floor, 1-4-1 Nihonbashi, Chuo-ku, Tokyo 1030027, Japan. Correspondence and requests for materials should be addressed to R.A. (email: agatar@jamstec.go.jp)

Post-earthquake deformation can be interpreted as a process of relaxing the stress perturbation caused by the earthquake rupture. It generally consists of the deformation due to continued, mostly aseismic slip on the megathrust (afterslip)[1] and viscoelastic relaxation in the asthenosphere[2]. Afterslip relaxes the stress perturbation by localized deformation in the region of the fault plane that surrounds the earthquake rupture. Viscoelastic flow relaxes the coseismic stress change by distributed, plastic deformation in the surrounding mantle[3,4]. The post-earthquake deformation of the 2011 $M_w$ 9.0 Tohoku-Oki earthquake was captured by a wide array of land-based[5,6] and seafloor[7–9] instruments. This widespread observation network captured a complex post-earthquake deformation field. Some near-trench seafloor stations moved seaward, in the opposite direction to the long-term subduction motion, while others moved landward (Fig. 1a). The post-earthquake vertical motion was also complex, with many seafloor stations moving in opposing directions than that on land. Several studies[7,8,10–12] claim that viscoelastic relaxation largely contributed to these patterns.

The 2011 $M_w$ 9.0 Tohoku-Oki earthquake induced a large stress perturbation in the surrounding lithosphere that accelerated the flow in the oceanic asthenosphere and in the mantle wedge. It is natural to expect that viscoelastic relaxation during the post-earthquake period can be described by the constitutive properties of peridotite, a rock assemblage of mostly pyroxene and olivine, under high temperature and pressure conditions[13]. Likewise, afterslip may be controlled by the frictional properties of the megathrust. Laboratory experiments suggest that the plastic deformation of mantle rocks is accommodated by a thermally activated flow that obeys a power-law relation between stress and strain rate[14,15]. The friction between the subducting slab and the upper plate is governed by a laboratory-derived kinematic friction law[16,17] that predicts the velocity of afterslip based on the stress evolution. Incorporating the laboratory-derived constitutive

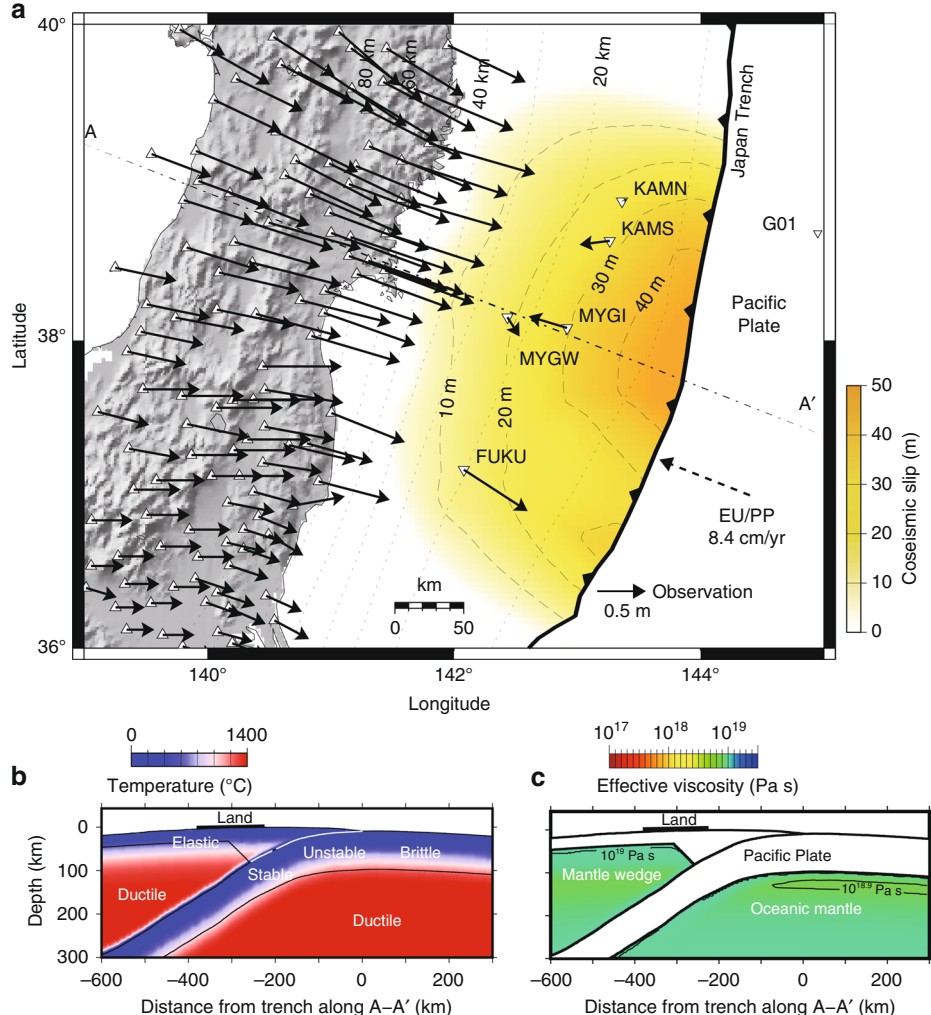

**Fig. 1** Post-earthquake deformation 2.8 years after the 2011 Tohoku-Oki Earthquake and surrounding material properties. **a** Measured displacement in the land stations (triangles) and the seafloor stations on both the continental plate and the pacific plate (inverse triangles). We removed some land stations for visibility. Coseismic displacement is not available in the station G01. Dashed-dotted and dotted lines are the location of the vertical cross-section (A–A′ profile) and the depth of the plate boundary, respectively. **b** Assumed temperature structure and frictional properties in the A–A′ profile. In the "unstable" region, where coseismic slip is input in our simulation, friction parameters are set as $-0.2 \leq A-B \leq -0.1$ MPa and $0.2 \leq L \leq 0.3$ m. In the "stable" region, where afterslip occurs in our simulation, $A-B = 0.1$ MPa and $L = 13$ m (also see Fig. 2b). The temperature values in the layers of elastic materials are not used in the simulation. **c** The assumed viscoelastic structure before the earthquake in the A–A′ profile. The mantle wedge and oceanic mantle are viscoelastic with $G_v = 65$ GPa. The remaining volume is elastic with $G_e = 45$ GPa. Poisson's ratio is $\nu = 0.25$ everywhere. The color indicates the effective viscosity in the Maxwell element before the earthquake. We used the same color scale as in Fig. 5 here to highlight the change due to the earthquake. Contribution from dislocation creep is dominant in the light green area, while viscosity in the linear term is dominant (see Methods) elsewhere

properties for viscoelastic flow and afterslip successfully explained the deformation that followed the 2012 $M_w$ 8.6 Indian Ocean earthquake[4], for which the surrounding rheological structure is rather simple. In contrast, most studies of the Tohoku-Oki earthquake employed simplified rheological models with linear viscoelastic flow in the mantle and kinematic afterslip[8,11,12,18–21], or explored more realistic rock properties in two-dimensional models[10,22]. This limitation of approach is due in part to the difficulty in dealing with the combination of the geometrical complexity and the nonlinear governing equations. Several of the linear viscoelastic models inferred from the Tohoku-Oki earthquake include a thin low-viscosity (weak) layer along the lithosphere–asthenosphere boundary (LAB) in the upper mantle[8,11,21]. A sharp decrease of seismic velocity at LAB[23,24] has been attributed to the presence of water or partial melts, which upholds the existence of a low-viscosity layer as a permanent rheology structure[8]. This interpretation remains controversial, as these findings require explanations consistent with mineral physics[14,15].

Here, we consider the three-dimensional response of the lithosphere–asthenosphere system following the 2011 $M_w$ 9.0 Tohoku-Oki earthquake with power-law viscoelastic flow in the mantle and afterslip on the megathrust, incorporating a realistic velocity structure for the Japanese margin, Earth's sphericity and laboratory-derived, nonlinear rock constitutive properties. We assume that the viscoelastic flow of the upper mantle is accommodated by steady-state dislocation creep, with the following stress–strain rate relationship[14]

$$\dot{\varepsilon}_M = A_M (C_{OH})^r \sigma^n \exp\left(-\frac{H}{RT}\right), \quad (1)$$

where $\varepsilon_M$ is the norm of the strain in the Maxwell element in a Burgers material (see Methods), $A_M$ is a pre-exponential factor, $C_{OH}$ and $r$ are the water concentration and its exponent, $\sigma$ is the norm of deviatoric stress tensor, $n$ is the stress exponent, $H = Q + p\Omega$ is the activation enthalpy, $R$ is the universal gas constant, and $T$ is the temperature. The enthalpy incorporates the activation energy $Q$ and the activation volume $\Omega$ and depends on the confining pressure $p$. In addition, we incorporate the transient creep that is thought to take place during the early stage of post-earthquake transients[4,25]. We use a model that includes the transient effect of dislocation creep[4], as

$$\dot{\varepsilon}_K = A_K (C_{OH})^r |\sigma - 2G_K \varepsilon_K|^n \exp\left(-\frac{H}{RT}\right), \quad (2)$$

where $\varepsilon_K$ is the norm of the strain in the Kelvin element in a Burgers material, $A_K$ is a pre-exponential factor and $G_K$ is a work hardening coefficient. Here we use the same parameters as in Eq. (1) with $A_K = A_M$ and $G_K = G$, where $G$ is rigidity. We combine dislocation creep with diffusion creep, but the latter does not play a significant role in our short-term simulations (see Methods). For the same reason, we did not include the transient effect of diffusion creep. We assume that the velocity of afterslip on the megathrust is governed by the rate-dependent and state-dependent friction, given by the constitutive law,

$$V = V_* \exp\left(\frac{\tau - (\tau_{s*} + \Delta\tau_s)}{A}\right), \quad (3)$$

combined with the aging law[17],

$$\Delta\dot{\tau}_s = \frac{B}{L/V_*} \exp\left(-\frac{\Delta\tau_s}{B}\right) - \frac{BV}{L}, \quad (4)$$

where $V$ is slip velocity, $V_*$ is the reference velocity, $\tau$ is the shear traction, $\tau_{s*}$ is the steady-state frictional resistance, and $\Delta\tau_s$ is a state variable analogous to the "strength as a threshold"[26]. $A$ is a

parameter that controls the fracture energy consumed during fault slip, the frictional parameter $B$ controls strength recovery, and $L$ controls the slip weakening distance. Simulating the dynamics of this nonlinear system in three-dimensions with realistic elastic, frictional, and viscoelastic properties requires state-of-the-art modeling strategies[27,28] (see Methods). Following this approach, we show the post-earthquake deformation in Tohoku to be caused by rapid flow in the asthenosphere, due to temporarily decreased viscosity because of coseismic stress.

## Results

**Cumulative 2.8 year post-earthquake displacement.** The temperature profile used in Eqs. (1) and (2) is based on a two-dimensional model for the Tohoku region[29], which we expanded along strike with a mantle temperature of 1380 °C (Fig. 1b), compatible with another study[4]. We converted the background shortening rate of $10^{-8}$ yr$^{-1}$ to determine the background stress based on the rheological law[30]. For the initial condition of the simulation, we borrow the coseismic slip (Fig. 1a) and the fault constitutive properties (i.e., $V$, $\tau$, $\Delta\tau_s$, $A$, $B$ and $L$) (Figs. 1, 2) from a simulation of giant earthquakes in the Tohoku region[31] (see Methods for details). We divide the region into three plates: a continental plate that includes the North-American and Eurasian plates and two oceanic plates, the Pacific and the Philippine Sea plates. Each tectonic plate consists of an elastic layer near the surface (the crust and the lithospheric mantle) and a viscoelastic mantle layer below (Figs. 1, 3). The elastic and viscoelastic layers in the three plates share the same elastic properties (Fig. 1c).

Our simulated deformation shows similar patterns to the observation data for the cumulative 2.8 year post-earthquake displacement in the horizontal direction (Fig. 4a) when we choose the following rock properties $K = 10^{0.56}$ MPa$^{-n}$ s$^{-1}$, $C_{OH} = 1000$ ppm H Si$^{-1}$, $Q = 430$ kJ mol$^{-1}$, $r = 1.2$, $\Omega = 13.5$ cm$^3$ mol$^{-1}$, and $n = 3$ (see Methods). For simplicity, we assumed a similar average water content in the oceanic asthenosphere and in the mantle wedge, even though water concentration may be larger in the mantle wedge corner due to slab dehydration[32]. The values adopted for the activation energy and the activation volume fall well within the uncertainties constrained by laboratory experiments[15], i.e., $Q = 410 \pm 50$ kJ mol$^{-1}$ and $\Omega = 11 \pm 3$ cm$^3$ mol$^{-1}$ for olivine, despite the required extrapolation to different temperature and pressure conditions. This indicates that the laboratory-derived rheological and frictional models with the proper in situ conditions allow us to make first-order predictions about how the lithosphere–asthenosphere system will deform in response to a large earthquake.

**Effective viscosity and time series of displacement.** The temporal and spatial evolution of effective viscosity after the giant earthquake naturally results from the nonlinear constitutive relations (1) and (2) and plays an important role in the rapid and complex deformation that occurs during the post-earthquake period[33]. In response to the large (above 1 MPa) stress perturbation in the upper mantle, the effective viscosity (see Methods for the definition) was largely reduced shortly after the earthquake in the depth of 100–200 km in the oceanic mantle and 80–180 km in the mantle wedge (Fig. 5). Temporal increase of effective viscosity explains well the time series of horizontal displacement in the station MYGI and some land stations that are aligned in the trench normal direction from the epicenter (Fig. 6). The misfit in the station MYGW is likely due to the dominance of the elastic response due afterslip there, which we discuss in the Discussion section.

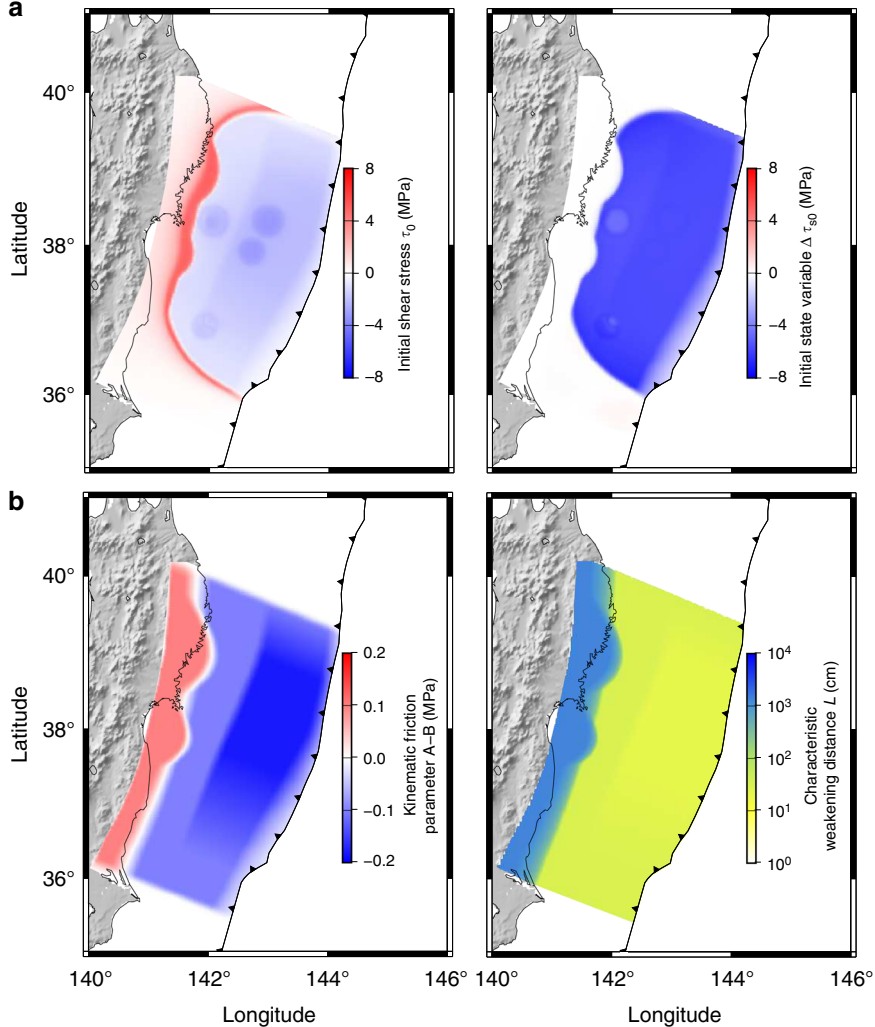

**Fig. 2** The variables and parameters that are taken over from a simulated $M_w$ 9 earthquake scenario produced by a past study. **a** Shear stress ($\tau$) and state variable ($\Delta\tau$) used as the initial values. The initial value of slip velocity ($V$) is calculated using these values with Eq. (3). **b** Frictional parameters. Afterslip occurs mainly in the area where $A - B$ is positive and $L$ is large

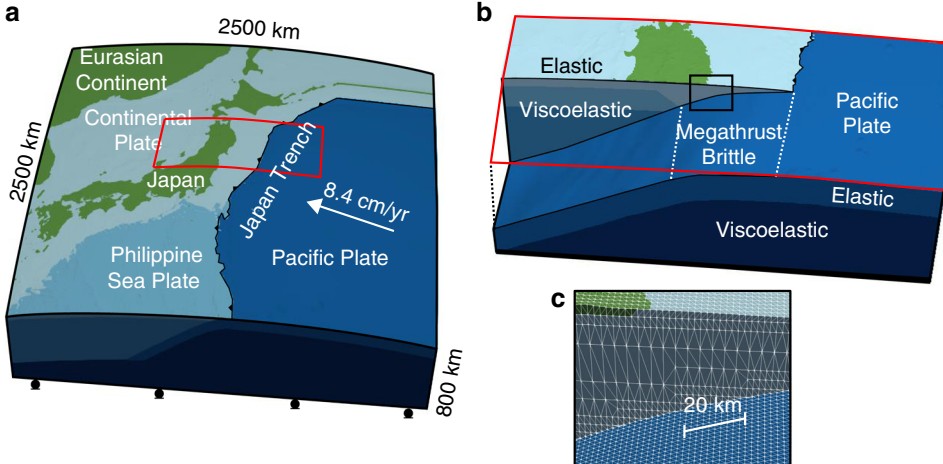

**Fig. 3** The finite-element model used in our study. **a** Overview, **b** close-up view for the region of the red rectangle in **a** with the location of the megathrust and **c** close-up view for the region of the black rectangle in **b** with finite-element mesh patterns. The elements with the same color are in the same structural component (we have six of them, elastic and viscoelastic layer in three plates). The green color is used to distinguish the elements that are located above sea level. The green elements have the same material properties as those in the continental plate

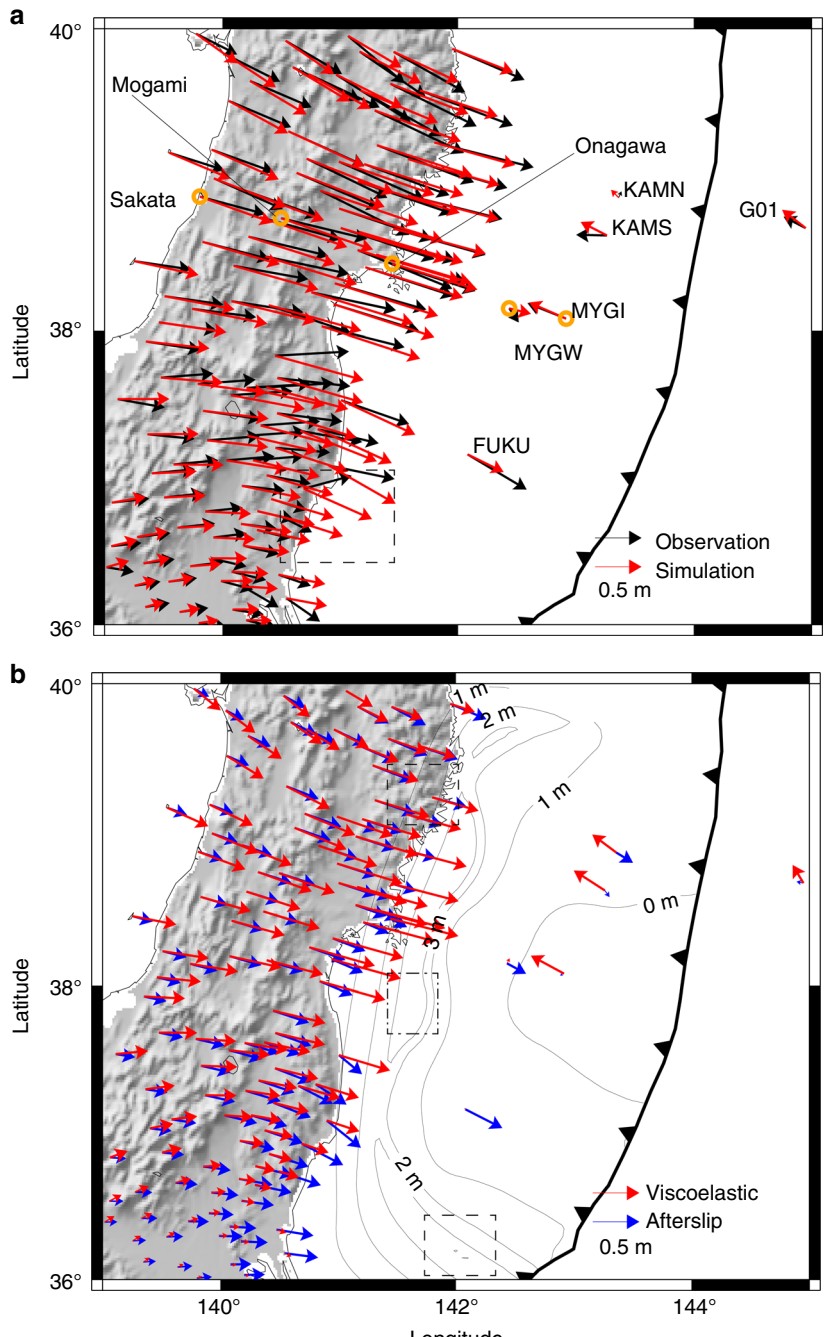

**Fig. 4** Post-earthquake deformation of the 2011 Tohoku-Oki earthquake. **a** The horizontal component of 2.8-year post-earthquake displacements. In the station G01, the contribution from the plate convergence rate (shown in Fig. 1a), which is not included in our simulation scheme, is added to the simulation result (see Methods). In addition, displacement in the period 1.5 years and 2.8 years after the earthquake is plotted in this station because of the limitation of data availability. Displacement time series in the stations marked by orange circles are shown in Fig. 6. **b** The horizontal components of 2.8-year post-earthquake displacements in the simulation broken down into the contribution from elastic deformation due to afterslip and viscoelastic flow. The viscoelastic component includes the contribution from both coseismic slip and afterslip. The contour lines indicate accumulated afterslip for 2.8 years. The fit to the horizontal displacements in the station FUKU would be better if large afterslip in the dashed rectangle were slightly closer to FUKU

**Decomposition of source mechanisms and induced deformation**. In our model, the post-earthquake displacements result from mechanically coupled afterslip and ductile flow. Both mechanisms are initially driven by the coseismic stress change, but they subsequently influence each other. Despite that coupling, the kinematics of deformation can be uniquely attributed to one source mechanism or the other: the displacements are a linear function of the slip and viscous strain distribution[34–37]. We exploit these relationships (see Methods) to unravel the relative

contributions of afterslip and viscoelastic flow within the subduction zone (Figs. 4, 7). The flow of low-viscosity mantle material below the trench axis drives westward motion around the trench, explaining the continued displacement of the seafloor stations located above the coseismic rupture (MYGI, KAMS, and KAMN, Fig. 4b). The accelerated flow in the mantle wedge contributes to the eastward displacement of GPS stations on land. Afterslip on the megathrust is essential to explaining the deformation on land, but also the spatial pattern of displacement of

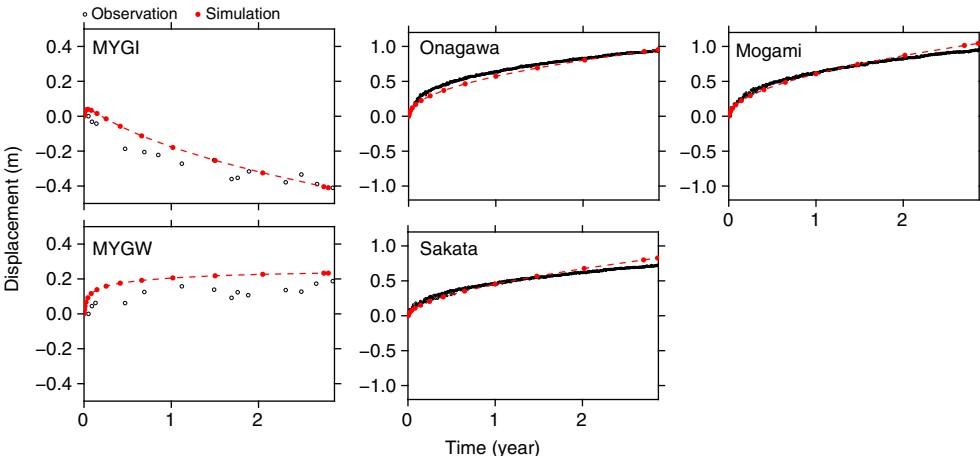

**Fig. 5** Distributions of effective viscosity. The panels show those in the steady-state and transient creep **a**, **b** shortly (at 0 year), **c**, **d** at 1 year and **e**, **f** at 2.8 years after the earthquake. See Methods for the definition of effective viscosity. The dashed line indicates summation of the background stress and the coseismic stress (norm of deviatoric stress tensor). Due to the power-law the stress relaxation is accompanied by material hardening, with a temporal increase in effective viscosity. As the material hardens, deformation is progressively accommodated by steady-state creep

**Fig. 6** The displacement time series in the trench-perpendicular direction at the stations denoted by the orange circles in Fig. 4a. Relatively large misfit in the station MYGW is discussed in the main text

the seafloor stations, such as eastward displacement seen in the stations FUKU and MYGW (Fig. 4b). Both these stations are in locations where viscoelastic flow produces little horizontal displacement, making the post-earthquake response due to the afterslip dominant there (Fig. 7).

**Discussion**

Remarkably, the spatial distribution of effective viscosity derived from laboratory data and coseismic stress change is similar to those inferred from optimization of simplified linear viscoelastic models[8,11,21]. The effective viscosity shortly after the earthquake

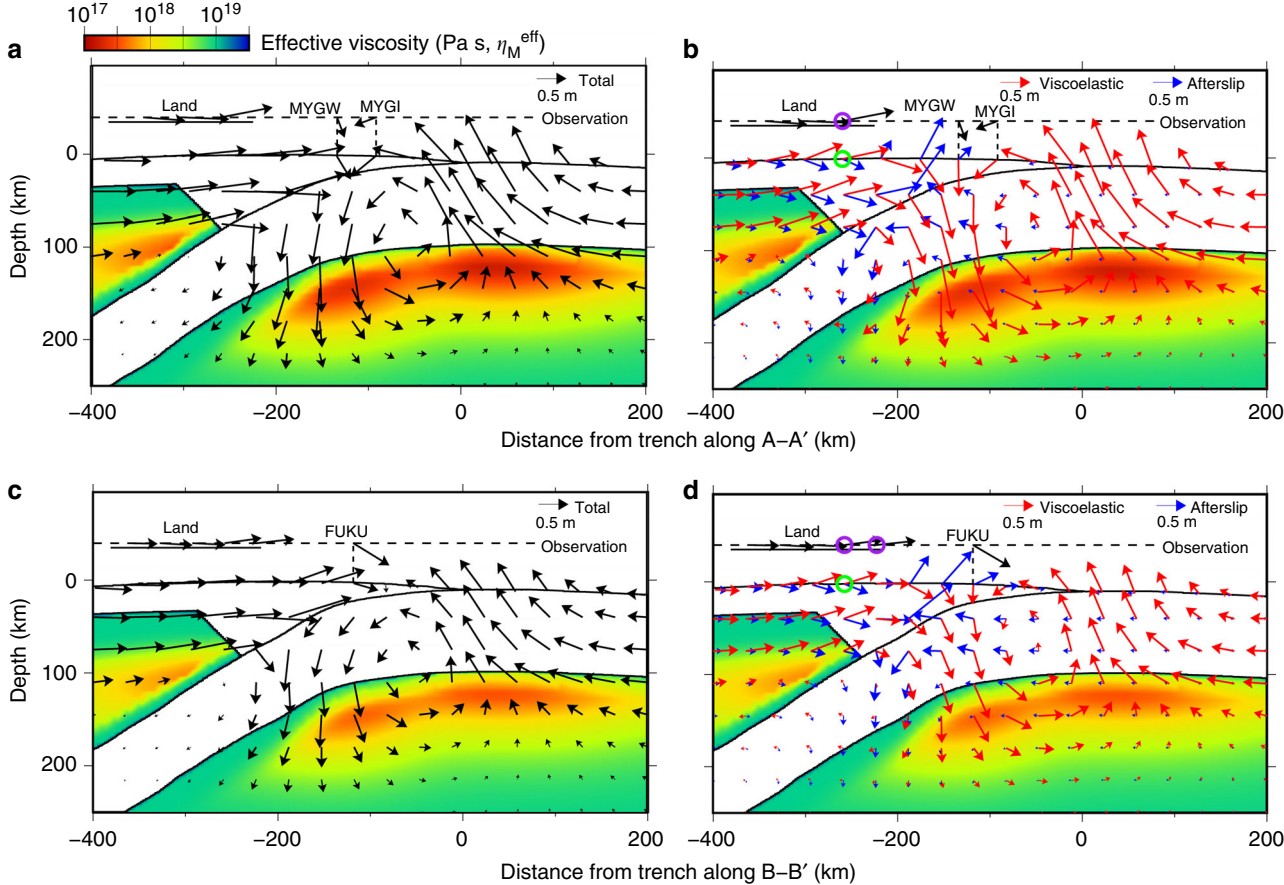

**Fig. 7** Simulation results on the vertical cross-sections parallel to the plate convergence direction, going through seafloor stations. **a** Displacement in the A–A' profile with MYGW and MYGI and **b** decomposition into contribution from viscoelastic relaxation and afterslip. **c** Displacement in the B–B' profile, which is parallel to A–A' and runs by the station FUKU and **d** decomposition into contribution from viscoelastic relaxation and afterslip. The figures on the left are for the total displacement after 2.8 years. The panels on the right show the contribution from elastic deformation due to afterslip and viscoelastic flow after 2.8 years. The color indicates the distribution of effective viscosity in the Maxwell element shortly after the earthquake. The black arrows on the horizontal dashed line are the observed displacements. In the location of purple circles, observation data shows uplift, while in the green circles, computed uplift viscoelastic displacement is canceled out by subsidence due to afterslip

is around $2 \times 10^{17}$ Pa s at the minimum both in the mantle wedge and the oceanic mantle. This is equivalent to the viscosity in a linear transient creep model that fits observed post-earthquake deformation during the early stage[8]. The LAB, originally identified as a low-seismic-velocity layer[23,24], has also been associated with a permanent low-viscosity structure. However, our result suggests that the LAB hosts a rapid mantle flow with temporarily decreased viscosity in response to large coseismic stress, rather than a permanent low-viscosity layer. A recent experimental study suggests that the presence of water, which has been invoked to explain a permanent low-viscosity structure at the LAB, is not compatible with the low seismic velocity[38]. Further studies are required to unravel the nature of the LAB.

Despite the excellent fit at numerous stations in the far-field, there remain a few discrepancies with the near-field data, presumably because our model does not include some fine details of the coseismic rupture offshore. For example, the simulated horizontal displacement at the station FUKU is nearly half of the measured one, despite a good agreement in the azimuthal direction. A peak of the amplitude of afterslip in the dashed rectangle in Fig. 4b should be slightly closer to station FUKU to better fit the data, perhaps indicating that the coseismic slip was overestimated in this region. Such afterslip distribution should also fit better the horizontal displacements in the southern part of the land area (the dashed rectangle in Fig. 4a). In addition, the displacement

time series in the station MYGW (Fig. 6) shows larger displacements in the plate convergence direction compared to the observed one. Figure 4b suggests that this is because the azimuthal direction of the elastic response due to the afterslip is almost parallel to the plate convergence direction, while the observation presents a displacement in the south-east direction. Smaller afterslip at the south of Onagawa (the dot-dashed rectangle in Fig. 4b), which is more consistent to the estimated afterslip distributions in previous studies[8,11], is likely to produce a displacement with a similar azimuthal direction to the observation. In the vertical displacement, significant uplift is observed in the fore-arc (the purple circles in Fig. 7).

In the trench-normal profile of the stations MYGI and MYGW, although viscoelastic flow in the simulation produces uplift in this region, subsidence due to afterslip cancels it out (the green circles in Fig. 7). A significant portion of this uplift in viscoelastic flow is due to stress change associated with afterslip, which we inferred from simulations of viscoelastic flow that exclude afterslip (the green circles in Fig. 8a). Without the interaction between afterslip and viscoelastic flow, the computed 2.8-year horizontal displacements are reduced by more than 10% in some of the land stations, and the vertical ones change by more than 30% in many stations in both the land and the seafloor (Fig. 8b). As afterslip in the near field can be highly sensitive to the details of the coseismic rupture, these residuals may be caused by still unresolved slip

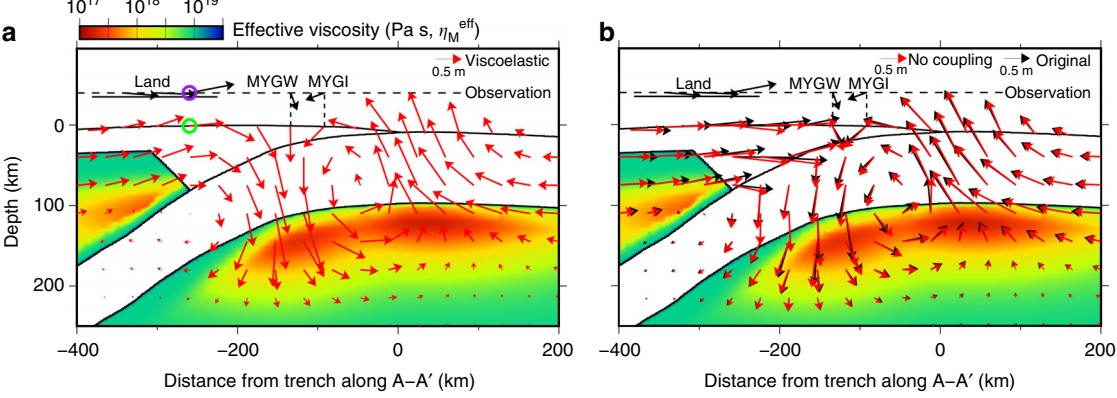

**Fig. 8** Displacement without the nonlinear interaction in the A–A′ profile. **a** Power-law viscoelastic flow in 2.8 years without considering afterslip in the vertical cross-section of the station MYGI and MYGW. In the green circle, uplift is significantly smaller than in the case with afterslip, shown in Fig. 7a. **b** Comparison between the total 2.8-year displacement in the original simulation (black, the same as "total" in Fig. 7) and the result without interaction between afterslip and viscoelastic flow (red). As a result, the computed horizontal displacements are reduced by more than 10% in some of the land stations, and the vertical ones change by more than 30% in many stations in both of the land and the seafloor. The color indicates the distribution of effective viscosity in the Maxwell element shortly after the earthquake

patterns of the mainshock. Nevertheless, our results highlight significant nonlinear interactions among coseismic slip, afterslip, and viscoelastic flow.

Our study demonstrates that a rheological model of the plate boundary based on independent geological and geophysical data can make realistic, first-order predictions of the transient response of the lithosphere following giant earthquakes. Complex post-earthquake deformation of a large subduction zone earthquake can be well explained by taking into account the laboratory-derived friction and viscoelastic flow laws in a three-dimensional structural model. The discrepancy between the simulation and the data, particularly in vertical motions and in some seafloor stations, should be reduced, in principle, by refined models of the coseismic rupture and the in situ conditions, such as initial stress, temperature, and confining pressure, properties that are usually only constrained for long time scales[29,39]. The approach is generally applicable to other ocean-continent subduction zones, implying that our understanding of viscoelastic properties and rocks friction may be detailed enough to predict the slow deformation of the lithosphere during the postseismic and interseismic periods.

## Methods

**Rheology model for upper mantle.** We use the Burgers-type rheology, where the strain due to steady-state creep and transient creep are in series:

$$\varepsilon_v = \varepsilon_M + \varepsilon_K, \tag{5}$$

where $\varepsilon_v$ is the viscous strain. In the Maxwell element, the strain rates for dislocation creep and diffusion creep add up, as

$$\dot{\varepsilon}_M = A_M(C_{OH})^r \sigma^n \exp\left(-\frac{Q+p\Omega}{RT}\right) + \frac{1}{2\eta_l}\sigma, \tag{6}$$

where $\eta_l$ is a constant viscosity for diffusion creep. The viscosity for diffusion creep is $10^{1-2}$ times larger than effective viscosity for dislocation creep shortly after earthquakes of $M_w$ 8.2 and 8.6[4], so the influence of diffusion creep is not expected to be very large in the 2.8 years deformation after the 2011 $M_w$ 9.0 Tohoku-Oki earthquake. We use $\eta_l = 1 \times 10^{19}$ Pa s for the whole of the region, which is nearly the average value of the viscosity structure estimated for steady-state 2D model around the Japan Trench[30]. In a tensor notation,

$$(\dot{\varepsilon}_M)_{ij} = A_K(C_{OH})^r \sigma^{n-1} \exp\left(-\frac{Q+p\Omega}{RT}\right)\sigma_{ij} + \frac{1}{2\eta_l}\sigma_{ij}. \tag{7}$$

We define the effective viscosity $\eta^{eff} = \sigma/2\dot{\varepsilon}$, thus

$$\eta_M^{eff} = \frac{\eta_p \eta_l}{\eta_p + \eta_l}, \tag{8}$$

where $\eta_M^{eff}$ is effective viscosity in the Maxwell element and

$$\eta_p = \frac{1}{2A_K(C_{OH})^r}\sigma^{-n+1}\exp\left(\frac{Q+p\Omega}{RT}\right). \tag{9}$$

In the same manner, we can write the transient dislocation creep (2) in the tensor notation as

$$(\dot{\varepsilon}_K)_{ij} = A_K(C_{OH})^r q^{n-1} \exp\left(-\frac{Q+p\Omega}{RT}\right)q_{ij}, \tag{10}$$

where $q_{ij} = \sigma_{ij} - 2G_K(\varepsilon_K)_{ij}$ and $q = (q_{kl}q_{kl})^{1/2}$. Then, the effective viscosity of the transient dislocation creep is

$$\eta_K^{eff} = \frac{1}{2A_K(C_{OH})^r}q^{-n+1}\exp\left(\frac{Q+p\Omega}{RT}\right), \tag{11}$$

where $\eta_K^{eff}$ is effective viscosity in the Kelvin element.

Our temperature pattern (Fig. 1b) in the elastic slab is significantly different from the reference thermal model[29] in that it keeps a low temperature even in the depth deeper than 200 km. However, the absolute temperature does not affect the simulation results significantly because the high pressure at these depths hardens the material. In the simulation, we use the values proposed from laboratory experiments[15] for K, r, and n, while Q and Ω were chosen within the error bar obtained in the same experiments, so that the computed displacement values are more consistent with the data. We set the $C_{OH}$ value as an average in the upper mantle. Further study on more detailed variation of measured displacement should require considering heterogeneous distribution of water content[4,40].

**Coseismic slip and fault friction setting.** To compute the postseismic deformation, we borrow the frictional properties assumed in the simulations of Nakata and colleagues[31]. The top of the subducting slab is modeled as a frictional interface loaded by the same tectonic forces that drive subduction. We assume the force balance

$$\dot{\tau}_i = F_i(\mathbf{v} - \mathbf{v}_{pl}, \dot{\varepsilon}_v) - \gamma \dot{V}_i \tag{12}$$

where $\tau_i$ and $V_i$ are shear stress and slip velocity on the $i$th FEM node on the fault. $V_i$ is in the direction opposite to the convergence rate (Fig. 1). $\mathbf{v}$ and $\mathbf{v}_{pl}$ are vectors whose components are $V_i$ and $(V_{pl})_i$, the plate convergence rate. Here, the difference between $\mathbf{v}$ and $\mathbf{v}_{pl}$ is the source of deformation based on the back slip model[41], which assumes that the steady-state subduction does not contribute to the deformation at the free surface in the hanging wall. It means that the calculated displacement at the foot wall does not include the contribution from the subduction motion either. $V_{pl} = 8.4$ cm yr$^{-1}$ is used for the whole region in this study. The second term introduces the effect of the seismic radiation damping[42]. We use $\gamma = 0.3G/2c$, which is used in Nakata et al.[31] to reproduce a shorter duration during the 2011 Tohoku-Oki earthquake[43], where G is the rigidity and c is the shear wave velocity. In many previous studies, the simulations have been carried out assuming an elastic homogeneous half-space, where $\dot{\varepsilon}_v = 0$. This makes $F_i$ a linear function of $v$ and enable $F_i$ to be discretized by the boundary integral equation method (BIEM). In this study, we evaluate $F_i$ directly by using the finite element method (see the next section), in which $F_i$ can be a function of both $\mathbf{v}$ and $\dot{\varepsilon}_v$, and arbitrary geometry and material heterogeneity can be considered. We carry out time integration of Eq. (12) and the equations for the rate-dependent and state-dependent

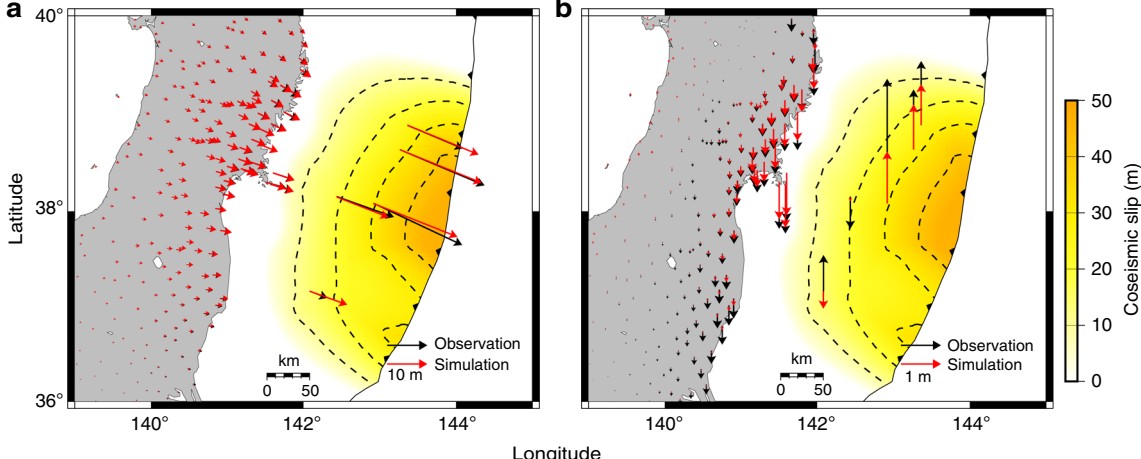

**Fig. 9** The input coseismic slip and comparison between computed and observed coseismic displacement including both the land and seafloor stations.
**a** Horizontal displacement. **b** Vertical displacement

friction law (3) and (4) using an adaptive time step fifth-order Runge–Kutta algorithm[44]. In our simulation, initial value of $\tau_i$ and $\Delta\tau_{si}$ is extracted from a time step right after the earthquake in the simulation of Nakata et al.[31] (Fig. 2a), multiplied by 0.7 to best-fit the geodetic data (Fig. 9). The initial value of $V_i$ is calculated with Eq. (3). Frictional parameters are also the same as in Nakata et al.[31], excluding that small patches for M7 earthquakes are removed (Fig. 2b). $A$ and $B$ values in Eqs. (3) and (4) are known to be normal-stress dependent: $A = a\sigma_n$ and $B = b\sigma_n$, where $\sigma_n$ is the normal stress. See Nakata et al. for the normal stress distribution. $V_*$ is set to be identical to $V_{pl}$.

Figure 9 shows the coseismic slip, the same as in Fig. 1, which we extracted from the cycle simulation results, and comparison between computed and observed coseismic displacement. Although this slip model is not inferred from observation data, it fits the horizontal component of coseismic crustal deformation data well when multiplied by 0.7. The stress distribution computed in response to this coseismic slip is used as the stress perturbation to compute power-law viscoelastic flow and afterslip evolution.

**Finite-element modeling**. In the finite-element modeling, we discretize the equations for viscoelastic deformation and fault friction using the mesh shown in Fig. 3. The mesh was constructed using an updated version of a meshing technique for quadratic tetrahedral elements based on a background structured grid[28]. In the method, at first a uniform background cell covering entire targeted domain was used, and it defined the resolution of the layer interfaces as $ds$. The geometries of the ground surface and interfaces were simplified slightly to maintain good element quality. At the same time, unnecessary elements were merged to generate larger elements elsewhere. This method enables automated and robust construction of high-resolution tetrahedral mesh directly from digital elevation model (DEM) data of crustal structure without creating a computer-aided design (CAD) model. The updated version of the meshing algorithm carries out an additional post process to minimize the simplification of the geometry in the ground surface and interfaces as much as possible. Input elevation data sets are based on 900 m resolution topography data (JTOPO30), the CAMP model[45] and a velocity data set for the Japanese Island[46]. From these data sets we constructed a finite element model in which the geometry of layer boundaries is in 2-km resolution ($ds = 2$ km) with slight modification. Using this finite element model, shear stress distribution on the fault, which is essential for computing stress-driven afterslip, is evaluated accurately in the target problem. The finite element mesh has 1,402,810,116 degree-of-freedom (DOF) and 346,885,129 tetrahedral elements. In viscoelastic material and elastic material, rigidity is $G_v = 65$ GPa and $G_e = 45$ GPa, respectively. Poisson's ratio is $\nu = 0.25$ and density is $\rho = 3300$ kg m$^{-3}$ everywhere, which setting follows Sun et al.[8]. Confining pressure is calculated as $p = \rho g z$, where $g$ is the gravitational acceleration and $z$ is depth.

To evaluate $F_i$ in Eq. (12), we applied an algorithm based on a viscoelastic finite element formulation[47,48], which we modified to consider nonlinear viscoelasticity. Slip velocity **v** is input to the finite-element model using the split node technique[49] to evaluate response displacement rate. We consider the effect of gravity using surface gravity approximation[50]. Since no inertia term is included in the equations, the problem is quasi-static, which ends up with solving an elliptic problem in every time step. It means we need to solve the system which has billions of DOF. We introduced a modified version[51] of a massively parallel FEM solver for computing crustal deformation[28] based on "GAMERA"[27] (a physics-based seismic wave amplification simulator, enhanced by a multiGrid method, Adaptive conjugate gradient method, Mixed precision arithmetic, Element-by-element method, and pRedictor by Adams–Bashforth method).

We run the calculation using 2048 computer nodes (16,384 computer cores) of the K computer at RIKEN Center for Computational Science[52], each computer node of which has one CPU (Fujitsu SPARC64 VIIIfx 8 core 2.0 GHz) and 16 GB of memory, for nearly 10 h to obtain the post-earthquake deformation for 2.8 years shown in Fig. 4.

**Geodetic data**. All the cumulative geodetic displacements plotted in the figures in this paper are adjusted to values relative to the stable part of the North American plate, on the basis of ITRF2005 model[53].

**Viscoelastic and afterslip contributions**. Figure 4b and the figures in the right in Fig. 6 present breakdown of computed displacement into contribution from elastic deformation due to afterslip and viscoelastic flow. In principle, calculated post-earthquake deformation in this study can be decomposed into elastic response due to cummulative afterslip and viscoelastic strain (e.g. refs. [34–37]). For example, in the case of the Maxwell-type rheology model for simplicity, $\mathbf{u}_{original}$, cumulative displacement vector at the GPS stations (corresponding to red arrows Fig. 4a), can be written as

$$\mathbf{u}_{original} = \mathbf{G}_d \Delta\mathbf{d} + \mathbf{G}_\varepsilon \Delta\boldsymbol{\varepsilon}_v, \qquad (13)$$

where $\Delta\mathbf{d}$ and $\Delta\boldsymbol{\varepsilon}_v$ are vectors for cumulative afterslip (corresponding to the black contour lines in Fig. 4b) and viscoelastic strain change, and $\mathbf{G}_d$ and $\mathbf{G}_\varepsilon$ are matrices for elastic Green's functions to map afterslip and viscoelastic strain change to displacement at the GPS stations. $\mathbf{u}_{afterslip} = \mathbf{G}_d \Delta\mathbf{d}$ and $\mathbf{u}_{viscoelastic} = \mathbf{G}_\varepsilon \Delta\boldsymbol{\varepsilon}_v$ correspond to the blue and red arrows in Fig. 4b, respectively. The second term of the right-hand side is more complex in the case of the Burgers-type rheology model, but the discussion here still applies. Note that $\Delta\mathbf{d}$ includes slip driven by coseismic stress, stress due to viscoelastic deformation and stress due to afterslip itself. In the same manner, $\Delta\boldsymbol{\varepsilon}_v$ includes strain change driven by coseismic stress, stress due to afterslip and stress due to viscoelastic relaxation itself. The contribution from each factor is nonlinearly coupled and cannot be decomposed from each other. $\mathbf{u}_{afterslip}$ and $\mathbf{u}_{viscoelastic}$ are calculated in the following three steps: 1. Extract accumulated 2.8 year afterslip distribution $\Delta\mathbf{d}$ that is computed based on the nonlinear interaction of the rate-dependent and state-dependent friction law and the nonlinear rock constitutive properties in the original simulation. 2. Compute elastic response displacement due to the cumulative after slip as $\mathbf{u}_{afterslip} = \mathbf{G}_d \Delta\mathbf{d}$ using the same finite-element model. 3. Compute the difference $\mathbf{u}_{viscoelastic} = \mathbf{u}_{original} - \mathbf{u}_{afterslip}$ to recover the contribution from viscoelastic flow.

We also present a result post-earthquake deformation simulation with "no interaction" between viscoelastic flow and afterslip (Fig. 8). In this simulation, we computed viscoelastic flow without the friction law (the red arrows in Fig. 8a), while computing afterslip without the nonlinear rock constitutive properties, only with pure elasticity. We finally summed up these to compute total deformation without their interaction (the red arrows in Fig. 8b).

**Code availability**. Computer codes for calculating viscoelastic relaxation and afterslip are available from the authors upon reasonable request.

## Data availability
GPS data are available from the Geospatial Information Authority of Japan (http://terras.gsi.go.jp/). Other relevant data in this work are available from the authors upon reasonable request.

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

## Acknowledgements

We used GPS data provided by the Geospatial Information Authority of Japan. This study was supported by Post K computer project (Priority issue 3: Development of Integrated Simulation Systems for Hazard and Disaster Induced by Earthquake and Tsunami) and JSPS Fellowship (26-8867). We obtained the results using the K computer at the RIKEN Center for Computational Science (Proposal number hp160221, hp170249,

and hp180207). S.B. was supported by the National Research Foundation (NRF) of Singapore under the NRF Fellowship scheme (National Research Fellow Awards Number NRF-NRFF2013-04), the Singapore Ministry of Education (AcRF Tier 1 grant RG181/16), and by the Earth Observatory of Singapore, under the Research Centres of Excellence initiative. Some figures were produced using GMT software, developed by P. Wessel and W.H.F. Smith.

## Author contributions

R.A. and T.H. designed and conducted the study. R.A, S.D.B. and T.H. wrote the manuscript. R.A., K.F. and T.Ichimura wrote the simulation code. S.D.B. and M.H. contributed to refining the simulation algorithm. S.D.B. and T.Iinuma contributed to the modeling. R.N. prepared the data used in the afterslip calculation.

## Additional information

**Competing interests:** The authors declare no competing interests.

