## [Peer Review File · Nature Communications]

Reviewers' comments:

Reviewer #1 (Remarks to the Author):

This work is another effort to model and explain postseismic deformation of the 2011 Tohoku-oki earthquake. Its main differences from previous efforts are the use of a nonlinear power-law rheology for the asthenospheric mantle and the use of a rate-state friction law to model afterslip. Because of the nonlinearity, the modeling is technically more challenging than previous works. However, because of the neglect of transient rheology, it is difficult to consider the work a significant scientific advance. I think the work is worth publishing, but with some major clarifications.

1. The paper claims that the steady-state power-law rheology is an improvement over the transient linear rheology in postseismic deformation modeling. This claim is incorrect and misleading. With uncertainty, one can argue that a nonlinear rheology may be an improvement over a linear rheology. However, with certainty, going from the transient rheology to the steady state rheology is a step backward. Transient creep is routinely observed in laboratory experiments, leading to the recognition that steady-state rheology is inappropriate for postseismic deformation modeling, as is also demonstrated by Freed et al. (2010, EPSL) using real postseismic data. Power-law flow indeed occurs in the lab and mantle. It may be useful to include it in the modeling, but not at the expense of ignoring the transient effect (i.e., throwing the baby out with the bathwater). There are efforts to introduce a transient-nonlinear law (e.g., Freed et al., 2012, JGR; Sobolev and Muldashev, 2017, G-cubed), although the assumptions involved are still to be studied. Given that the issue is not fully resolved, it is ok if the authors insist on applying a steady-state law to a transient process such as postseismic deformation, but they should present an unbiased discussion and to recognize the potential errors caused by ignoring the transient rheology.

2. The effective viscosity in the mantle wedge is quite high in the model (Fig. 7). Only in a very small area is it comparable to the transient viscosity used by Sun et al. (2015, Nature) and the Maxwell viscosity used by Freed et al. (2017, EPSL). One reason is that the authors assume the same viscosities for the continental and oceanic side of the system, ignoring the effects of high fluid contents in the mantle wedge. The resultant error can be large. The other reason is the neglect of the transient rheology, as discussed above. The authors emphasize that high strain rates right after the earthquake lead to low effective viscosities because of the power-law behaviour, similar to having a low viscosity in the phase of transient creep. This comparison is not quite right. With the transient rheology, the initial viscosity is low everywhere in the mantle wedge regardless of strain rates; with the steady-state power law, the effective viscosity is low only in very limited high-strain-rate areas. The consequence of a high mantle wedge viscosity is that much afterslip is needed downdip of the rupture zone in order to explain the fast seaward motion of the GPS sites on land. Most viscoelastic models for this earthquake with proper Earth structure do not feature much afterslip directly downdip of the rupture zone (see review by Wang et al., 2018, Geosphere, STB-2 volume).

3. The authors quote Cline II et al. (2018) to argue that a permanent low-viscosity LAB is not needed. Cline II et al. actually discussed the relevance of water presence in explaining anomalous seismic wave speed. The presence of water is only one proposed mechanism to explain the seismically inferred LAB. A more widely assumed mechanism is the presence of melts. I think there is still a lot to study and learn about the nature of the LAB, and the discussion in this paper should be more balanced.

4. Given the high nonlinearity of the model used in this work, it seems incorrect to decompose the contributions from afterslip and viscoelastic relaxation as discussed in section 0.4 (p. 10) and shown in Fig. 3. Neither is it correct to add the contribution from plate convergence after the simulation (stated in Fig. 3 caption). If the authors choose not to redo all the simulations, they

should in some way demonstrate that the errors are very small (by designing some simple tests?).

5. If I understand the presentation correctly, the authors did not try to explain GPS time series but only tried to fit the 2.8-year cumulative postseismic deformation. Please confirm if this is indeed the case. If so, can you say the new model is better than some of the published models that explain the continuous evolution of GPS displacements?

6. Other comments:

Page 3, paragraph 2. I do not understand the logic of the last sentence. What is said in the rest of the paragraph by no means indicates that the physical setting of the Japan subduction zone is "understood well enough to make accurate predictions about ...". The validity of applying the lab-derived rheological and frictional parameters of to in situ conditions is not known. Many trade-offs are involved. What we can reasonably hope is to develop an understanding of the first-order physical process by making various assumptions and simplifications.

Page 3, paragraph 4. The effective viscosity in the mantle wedge is very low only in a small region and is not remarkable. Also, the word "simplified" before "linear viscoelastic models" is redundant (see comment 1 above). The power-law model is also a simplified one (by ignoring transient creep).

Page 3, paragraph 4, last sentence. What the authors mean to say is "... recent finding that the seismically observed low-velocity boundary cannot be explained by the presence of water." However, what about melts? See comment 3 above.

Fig. 3 caption. The last part of the last sentence is difficult to understand.

Page 10, paragraph 1. For the purpose of this work, the 900 m resolution topography is obviously unnecessary. It is rather intriguing to see the co-existence (and gross incompatibility) of very fine details like this and the greatest simplifications such as the rather uniform elastic moduli and simple plate structure.

Page 10. Using the elastic response to $u_{\text{afterslip}}$ to represent the effect of afterslip in a nonlinear viscoelastic model is almost certainly incorrect, at least in theory (see comment 4 above).

Reviewer #2 (Remarks to the Author):

The work presented is computationally challenging and quite sophisticated. It is an impressive study. No other study has tried to model postseismic deformation with such a nearly complete physical model of coupled afterslip and nonlinear power-law mantle rheology. The authors show that they can match some of the first-order features of postseismic displacements recorded with geodesy. Unlike previous studies, the authors did not impose a viscosity structure to fit the data – they used rheology consistent with laboratory measurements and the thermal model to allow the viscosity structure to evolve naturally. This is a very important improvement over previous studies.

The work here is difficult and they have done a nice job, but I am left wondering why they do not show deformation over time. The authors do not show time series data and fits. They also do not clearly present the fit to vertical displacements. My experience is the vertical motions and time series data are very important for distinguishing between mechanisms.

The paper is clearly written and easy to follow.

I think this is an important contribution and should definitely be published. This paper will gain a lot of attention among people working on earthquake cycle deformation.

A few specific comments:

1. The initial conditions were kind of brushed off in this manuscript as a secondary issue. How important are initial stress and viscosity on postseismic deformation? To be clear, it seems you assume an initial uniform strain rate to get viscosity and stress, is this correct? Given that the earthquake cycle is not a steady process, is this an appropriate condition for the initial state? I don't think you ever said what the initial conditions are on the stable friction sliding zone.
2. The text in your Figure 1 caption is difficult to relate to equations in the main text. In the caption you discuss A-B and L, but these do not appear in equation (2), nor are they every defined in the text as far as I can tell. These parameters do appear in section 0.2 (Methods). Perhaps equations (8) and (9) belong in the main text. Do you use the same A-B and L parameters in your afterslip simulations as well as the cycle simulations described in section 0.2? This all deserves some clarification, I think.
3. In Figure 3b you show afterslip as contours. It looks like this overlaps coseismic rupture. Do you allow overlap? Is the overlap possible in velocity-weakening coseismic slip region? It would be helpful to have a figure somewhere (maybe in methods) that shows the distribution of weakening and strengthening regions in map view on the interface.
4. You fit the 2.8 years of cumulative horizontal displacements reasonably well. What about the vertical? It would be nice to see a map-view plot of the fit to vertical. I suspect, from my own experience, that the vertical is an important constraint.
5. Is there a reason you do not show time series data and fits? I suspect fit to time series is very illuminating about how well this model captures the deformation.

Answer from the authors to Reviewer #1

>This work is another effort to model and explain postseismic deformation of the 2011 Tohoku-oki earthquake. Its main differences from previous efforts are the use of a nonlinear power-law rheology for the asthenospheric mantle and the use of a rate-state friction law to model afterslip. Because of the nonlinearity, the modeling is technically more challenging than previous works. However, because of the neglect of transient rheology, it is difficult to consider the work a significant scientific advance. I think the work is worth publishing, but with some major clarifications.

Thank you very much for your fair and constructive comments on our work. Ignorance of transient creep in our model was the direct reason why we did not show the time series of displacement. We did not include the transient creep simply because there are still some uncertainties in how to model it.

In the new manuscript, we show the new simulation results incorporating transient creep based on one of the previous studies. This allows us to show the time series of displacement in a few observation stations. Please see the followings for the details.

Here L denotes the line number.

>1. The paper claims that the steady-state power-law rheology is an improvement over the transient linear rheology in postseismic deformation modeling. This claim is incorrect and misleading. With uncertainty, one can argue that a nonlinear rheology may be an improvement over a linear rheology. However, with certainty, going from the transient rheology to the steady state rheology is a step backward. Transient creep is routinely observed in laboratory experiments, leading to the recognition that steady-state rheology is inappropriate for postseismic deformation modeling, as is also demonstrated by Freed et al. (2010, EPSL) using real postseismic data. Power-law flow indeed occurs in the lab and mantle. It may be useful to include it in the modeling, but not at the expense of ignoring the transient effect (i.e., throwing the baby out with the bathwater). There are efforts to introduce a transient-nonlinear law (e.g., Freed et al., 2012, JGR; Sobolev and Muldashev, 2017, G-cubed), although the assumptions involved are still to be studied. Given that the issue is not fully resolved, it is ok if the authors insist on applying a steady-state law to a transient process such as postseismic deformation, but they should present an unbiased discussion and to recognize the potential errors caused by ignoring the transient rheology.

In the original manuscript, we did not consider transient-nonlinear creep because its behavior is still not fully understood as pointed out by Masuti et al. (2016). We wanted to focus on the relation between the 2.8-year cumulative post-earthquake deformation and rapid flow in the mantle with coseismically decreased viscosity by using only the steady-state laws. This simplification is justified by Figure 8 in Sobolev and Muldashev (2017), which implies that introduction of transient creep to deformation calculation using steady-state creep does not make so much difference after several months.

However, as you say, it is true that many recent studies discuss the importance of transient rheology in post-earthquake deformation (in addition to the studies you mentioned, Sun et al (2014), Masuti et al. (2016) and so on). Since our simulation code has been capable of it, for this revision we redid the numerical simulation introducing nonlinear transient creep with the constitutive relation and the parameters proposed by Masuti et al. (2016). Equation 2 describes the details. This also enables us to present the time series of displacement in a few observation stations. As shown below, introduction of transient creep seems to improve the fit of the displacement in the early stage in the land station (Please see the main text for the locations of the stations). In the manuscript, only the results with the transient creep is shown in Figure 7.

Discussion on the time series is added at L102 as the following:

“Temporal increase of effective viscosity takes place in the relaxation process of coseismic stress (Fig. ¥, ¥ref{fig:viscolater}), which explains well the time series of horizontal displacement in the station MYGI and some land stations that are aligned in the trench normal direction from the epicenter (Figure ¥, ¥ref{fig:timehistory}).

The misfit in the station MYGW is likely due to the dominance of the elastic response due afterslip there, which we discuss below.”

Reference:

Masuti, S., Barbot, S. D., Karato, S., Feng, L. & Banerjee, P. Upper-mantle water stratification inferred from observations of the 2012 Indian Ocean earthquake. *Nature* 538, 373-377 (2016).

>2. The effective viscosity in the mantle wedge is quite high in the model (Fig. 7). Only in a very small area is it comparable to the transient viscosity used by Sun et al. (2015, *Nature*) and the Maxwell viscosity used by Freed et al. (2017, *EPSL*). One reason is that the authors assume the same viscosities for the continental and oceanic side of the system, ignoring the effects of high fluid contents in the mantle wedge. The resultant error can be large. The other reason is the neglect of the transient rheology, as discussed above. The authors emphasize that high strain rates right after the earthquake lead to low effective viscosities because of the power-law behaviour, similar to having a low viscosity in the phase of transient creep. This comparison is not quite right. With the transient rheology, the initial viscosity is low everywhere in the mantle wedge regardless of strain rates; with the steady-state power law, the effective viscosity is low only in very limited high-strain-rate areas. The consequence of a high mantle wedge viscosity is that much afterslip is needed downdip of the rupture zone in order to explain the fast seaward motion of the GPS sites on land. Most viscoelastic models for this earthquake with proper Earth structure do not feature much afterslip directly downdip of the rupture zone (see review by Wang et al., 2018, *Geosphere*, STB-2 volume).

Two figures of the effective viscosity distribution shown in Figure 7 in the original manuscript (now Figure 5) are for 1-year and 2.8-year snapshot, respectively. The effective viscosity shortly after the earthquake newly shown in Figure 5 (the same effective viscosity distribution as shown in Figure 6) clearly shows that low viscosity in mantle wedge is seen in a quite large volume, which includes horizontally the whole of the Japanese Islands in the trench normal direction.

Still, it may be true that afterslip calculated in the simulation is too large, e.g., the area marked by the pink circle in Figure 4b. Smaller afterslip there is likely to improve the fit in the station MYGW. This can be compensated by a larger water amount in the mantle wedge, for which we currently simply set a homogeneous distribution with $\text{COH}=1000\text{ppmH/Si}$. We added the following sentences in L125:

“In addition, the displacement time series in the station MYGW (Figure ¥,¥ref{fig:timehistory}}) shows larger displacements in the plate convergence direction compared to the observed one. Figure ¥,¥ref{fig:disp}b suggests that this is because the azimuthal direction of the elastic response due to the afterslip is almost parallel to the plate convergence direction, while the observation presents a displacement in the south-east direction.

Smaller afterslip at the south of Sendai (the dot-dashed rectangle in Figure ¥,¥ref{fig:disp}b),

which is more consistent to the estimated afterslip distributions in previous studies (Sun 2014, Freed 2017), is likely to produce a displacement with a similar azimuthal direction to the observation.”

>3. The authors quote Cline II et al. (2018) to argue that a permanent low-viscosity LAB is not needed. Cline II et al. actually discussed the relevance of water presence in explaining anomalous seismic wave speed. The presence of water is only one proposed mechanism to explain the seismically inferred LAB. A more widely assumed mechanism is the presence of melts. I think there is still a lot to study and learn about the nature of the LAB, and the discussion in this paper should be more balanced.

As you say, the presence of water is just one proposed mechanism. We do not intend to say that power-law creep is the only one possible mechanism of low-viscosity layer, but our message is that further studies are required for learning about the nature of the LAB. We modified the text and clarified this point at L115 as the following:

“A recent experimental study suggests that the presence of water, which has been invoked to explain a permanent low-viscosity structure at the LAB, is not compatible with the low seismic velocity (Cline II 2018). Further studies are required to unravel the nature of the LAB.”

>4. Given the high nonlinearity of the model used in this work, it seems incorrect to decompose the contributions from afterslip and viscoelastic relaxation as discussed in section 0.4 (p. 10) and shown in Fig. 3.

Since we simulate two way coupling of frictional slip and nonlinear viscoelastic relaxation, the calculated afterslip necessarily includes effect by viscoelasticity. In this sense, as you pointed out, we cannot decompose the contributions from afterslip and viscoelastic relaxation completely. What Section 0.4 describes is decomposition of elastic response due to afterslip and viscoelastic relaxation, which should be theoretically correct. We clarified it at the captions of Figure 4 and 6, and L230 in the new manuscript.

Viscoelastic relaxation can be considered as elastic response due to summation of elastic response due to viscoelastic strain change in each finite element, which allows this direct decomposition. Qiu et al, Nature Communications (2018) carried out simultaneous geodetic inversion of afterslip and mantle wedge viscosity from post-earthquake deformation data based on this relation. At L239, we added the sentence below

“Such decomposition is valid because $\mathbf{u}_{\text{viscoelastic}}$ is identical to the summation of elastic response due to viscoelastic strain change in each finite domain that discretizes the mantle” (Qiu2018).”

Reference:

Qiu, Q., Moore, J. D., Barbot, S., Feng, L., & Hill, E. M. (2018). Transient rheology of the Sumatran mantle wedge revealed by a decade of great earthquakes. *Nature communications*, 9(1), 995.

>Neither is it correct to add the contribution from plate convergence after the simulation (stated in Fig. 3 caption).

We first would like to emphasize that such treatment is only limited to the station on the Pacific Plate (i.e. G01 station). The simulation is based on the back slip model (Savage 1983), which assumes that steady state subduction (i.e. $V=V_{pl}$) does not contribute to the deformation at the free surface in the hanging wall (this is why the plate convergence rate is subtracted from the slip velocity in Equation (12));

However, this model requires adding the contribution from the subduction to the calculated displacement on the foot wall (the oceanic plate) because the model does not introduce the boundary condition for plate motion. Since G01 is on the foot wall, we need to carry out this post-process on its displacement.

(An example: in the back slip model, suppose that $V=V_{pl}$ holds everywhere in the fault, there is no source and therefore no deformation at any place. However, in reality, the stations in the foot wall is surely moving toward the trench at the plate convergence velocity even such an ideal situation. To fix this discrepancy, we need to add the contribution from plate convergence as a post process)

We added the following sentence four lines below L176:

“Here the difference between \mathbf{V} and \mathbf{V}_{pl} is the source of deformation based on the back slip model (Savage1983), which assumes that the steady state subduction does not contribute to the deformation at the free surface in the hanging wall. It means that the calculated displacement at the foot wall does not include the contribution from the subduction motion either.”

> If the authors choose not to redo all the simulations, they should in some way demonstrate that the errors are very small (by designing some simple tests?).

I do not fully understand what you intended to say. What kind of error do you mean?

>5. If I understand the presentation correctly, the authors did not try to explain GPS time series but only tried to fit the 2.8-year cumulative postseismic deformation. Please confirm if this is indeed the case. If so, can you say the new model is better than some of the published models that explain the continuous evolution of GPS displacements?

In the original manuscript, no, we did not consider GPS time series because we did not think it is proper to discuss time series without introducing transient creep. Now we redid the simulation with nonlinear transient creep, which allows to show the time series of displacement in Figure 7. Discussion on the time series is added at L102 as the following:

“Temporal increase of effective viscosity takes place in the relaxation process of coseismic stress (Fig. ¥,¥ref{fig:evicolater}}), which explains well the time series of horizontal displacement in the station MYGI and some land stations that are aligned in the trench normal direction from the epicenter (Figure ¥,¥ref{fig:timehistory}}).

The misfit in the station MYGW is likely due to the dominance of the elastic response due afterslip there, which we discuss below.”

Relating to this comment, we would like to emphasize that we do not intend to improve the fit from the results in other studies. We incorporate much smaller number of parameters in rheological and slip models, seeking to let laboratory-derived physical models explain the data as much as possible. This is in contrast to the previous studies such as Sun et al. (2014) and Freed et al (2017) estimating viscosity in several layers in mantle and slip in fault patches to fit the data. These two studies first tried to fit the data as much as possible using simple models and then tried to discuss the underlying physics. Therefore, our study and these two tackled the same problem from opposite directions (that is why it is interesting to see the similar effective viscosity distribution in our study and theirs), in which situation we do not mean to discuss too much on comparison of their fitness to the data.

For this reason, it is also intrinsically natural that our model might fit measured displacement in some locations slightly worse than the previous studies.

>6. Other comments:

Page 3, paragraph 2. I do not understand the logic of the last sentence. What is said in the rest of the paragraph by no means indicates that the physical setting of the Japan subduction zone is “understood well enough to make accurate predictions about ...”. The validity of applying the lab-derived rheological and frictional parameters of to in situ conditions is not known. Many trade-offs are involved. What we can reasonably hope is to develop an

understanding of the first-order physical process by making various assumptions and simplifications.

What you point out is true, and we think that such studies as this work and Masuti et al. (2016), which suggests that lab-derived rheological models explain well the post-earthquake deformation after the 2012 Indian Ocean earthquake, should be continuously done to test the validity in the future. In this sense, a more moderate way of saying is more suitable here. We modified it at L86 as the following:

“This indicates that the laboratory-derived rheological and frictional models with the proper in-situ conditions allow us to make first-order predictions about how the lithosphere-asthenosphere system will deform in response to a large earthquake.”

>Page 3, paragraph 4. The effective viscosity in the mantle wedge is very low only in a small region and is not remarkable. Also, the word “simplified” before “linear viscoelastic models” is redundant (see comment 1 above). The power-law model is also a simplified one (by ignoring transient creep).

A larger view of distribution of effective viscosity shortly after the earthquake newly shown in Figure 5 clearly suggests that low viscosity in mantle wedge is seen in a quite large volume, which horizontally covers the Japanese Island along the trench normal direction. In the new simulation, we introduced transient creep, which should justify the usage of “simplified” here.

>Page 3, paragraph 4, last sentence. What the authors mean to say is “... recent finding that the seismically observed low-velocity boundary cannot be explained by the presence of water.” However, what about melts? See comment 3 above.

We newly mention partial melt as another possible mechanism at L52. In addition, we did not conclude anything but just raised up discussion stating that

“A recent experimental study suggests that the presence of water, which has been invoked to explain a permanent low-viscosity structure at the LAB, is not compatible with the low seismic velocity (Clineil2018). Further studies are required to unravel the nature of the LAB.”

at L115.

>Fig. 3 caption. The last part of the last sentence is difficult to understand.

What we meant is that large afterslip in the green ellipse seems to be the source of bad fit in FUKU. We modified the sentence in the caption as follows.

“The fit to the horizontal displacements in the station FUKU would be better if large afterslip in the green ellipse were slightly closer to FUKU.”

>Page 10, paragraph 1. For the purpose of this work, the 900 m resolution topography is obviously unnecessary. It is rather intriguing to see the co-existence (and gross incompatibility) of very fine details like this and the greatest simplifications such as the rather uniform elastic moduli and simple plate structure.

The resolution of the input topography data is 900 m, but that of the FE model is 2km. We emphasized it at L203 in the new manuscript.

In numerical simulation, a process known as “verification and validation (V&V)” is essential. Verification is a process that checks whether a numerical model assumed for a certain physical phenomenon (e.g., a partial differential equation) is solved correctly by the simulation code. After some preliminary runs for the verification process, we have confirmed that 2km resolution is necessary and sufficient for this simulation. Validation is a process that checks whether the numerical model reasonably explains the target physical phenomenon. Simplification in elastic moduli and the plate structure is a matter of validation. This classification avoids confusion when discussing differences between the results from numerical solutions and the actual observation data. Hence in our point of view, it is rather natural to see the co-existence of fine resolution topography and simplification of the structure.

Reference:

Guide for verification and validation in computational solid mechanics, The American Society of Mechanical Engineers, 2006.

>Page 10. Using the elastic response to $u_{\text{afterslip}}$ to represent the effect of afterslip in a nonlinear viscoelastic model is almost certainly incorrect, at least in theory (see comment 4 above).

As answered to the comment 4, it is correct to say “decomposition of the contribution from elastic deformation due to afterslip and viscoelastic relaxation”. Viscoelastic relaxation can be considered as elastic response due to superposition of elastic response due to viscoelastic strain change in each finite element, which allows this direct decomposition.

Answer from the authors to Reviewer #2

>The work presented is computationally challenging and quite sophisticated. It is an impressive study. No other study has tried to model postseismic deformation with such a nearly complete physical model of coupled afterslip and nonlinear power-law mantle rheology. The authors show that they can match some of the first-order features of postseismic displacements recorded with geodesy. Unlike previous studies, the authors did not impose a viscosity structure to fit the data – they used rheology consistent with laboratory measurements and the thermal model to allow the viscosity structure to evolve naturally. This is a very important improvement over previous studies.

The work here is difficult and they have done a nice job, but I am left wondering why they do not show deformation over time. The authors do not show time series data and fits. They also do not clearly present the fit to vertical displacements. My experience is the vertical motions and time series data are very important for distinguishing between mechanisms.

The paper is clearly written and easy to follow.

I think this is an important contribution and should definitely be published. This paper will gain a lot of attention among people working on earthquake cycle deformation.

Thank you very much for your encouraging and constructive comments. As written below, we originally did not show the time series because we did not incorporate transient creep in the simulation. In the new manuscript, we show the new simulation results incorporating transient creep, which allows us to show the time series of displacement in a few observation stations.

Here L denotes the line number.

>1. The initial conditions were kind of brushed off in this manuscript as a secondary issue. How important are initial stress and viscosity on postseismic deformation? To be clear, it seems you assume an initial uniform strain rate to get viscosity and stress, is this correct? Given that the earthquake cycle is not a steady process, is this an appropriate condition for the initial state? I don't think you ever said what the initial conditions are on the stable friction sliding zone.

When we discuss the initial condition (stress) in our simulation, we would like to distinguish two kinds: One is for the bulk rheology in the mantle, and the other is for the fault plane (the friction sliding zone). For the bulk rheology, assumption of an initial uniform strain rate may not be the best, but at least we believe is not too bad. On the other hand, as you point out, it

is clearly not suitable for the initial state of earthquake cycle calculation. So instead, we took over a result from an earthquake cycle simulation in Tohoku region carried out by Nakata et al. (2016). Because initial stress and state variables on the fault plane were computed in the simulation, we used them as the initial conditions on the fault plane. Although I described in Method part, it is important to mention it in the main text. Therefore, I moved the figures for the initial conditions in Figure 2. I emphasize this point as well in L68 as below:

“For the initial condition of the simulation, we borrow the coseismic slip (Fig. \ref{fig:overview}a), the fault constitutive properties (i.e., V , τ , $\Delta\tau_{\text{rms}}$, A , B and L) (Fig. \ref{fig:overview}c and Fig. \ref{fig:friction}) from a simulation of giant earthquakes in the Tohoku region~\cite{Nakata2016} (see Methods for details).”

>2. The text in your Figure 1 caption is difficult to relate to equations in the main text. In the caption you discuss A-B and L, but these do not appear in equation (2), nor are they every defined in the text as far as I can tell. These parameters do appear in section 0.2 (Methods). Perhaps equations (8) and (9) belong in the main text. Do you use the same A-B and L parameters in your afterslip simulations as well as the cycle simulations described in section 0.2? This all deserves some clarification, I think.

As you guess, we have another equation besides Equation (2) for the fault friction law, which was described in Method part in the original manuscript. A-B and L appeared in it. Now I moved the equation to the main text (Equation 4) with explanation of the parameters as below at L67,

“the frictional parameter B controls strength recovery, while L controls slip weakening.”, so that it is easy relate the Figure 1 caption and the equations. The same A-B and L parameters as in the cycle simulations described in section 0.2 are used in this study. We moved the two-dimensional views of fault parameters from Method to the main text as well (Figure 2).

>3. In Figure 3b you show afterslip as contours. It looks like this overlaps coseismic rupture. Do you allow overlap? Is the overlap possible in velocity-weakening coseismic slip region? It would be helpful to have a figure somewhere (maybe in methods) that shows the distribution of weakening and strengthening regions in map view on the interface.

We do allow overlap. Because in our simulation we solve the velocity-weakening and strengthening region together, some remnant slip in the velocity-weakening region is counted

as afterslip (it is a matter of when to define the end of the coseismic slip). We moved the two-dimensional views of fault parameters from Method to the main text as well (Figure 2), which shows the spatial relation of the velocity-weakening and strengthening region with the caption as below:

“Afterslip occurs mainly in the area where $A-B$ is positive.”

>4. You fit the 2.8 years of cumulative horizontal displacements reasonably well. What about the vertical? It would be nice to see a map-view plot of the fit to vertical. I suspect, from my own experience, that the vertical is an important constraint.

We focus on the horizontal because it already provides large constraints for overall behavior of post-earthquake deformation due to viscoelastic relaxation and afterslip. Generally speaking, reproducing observed vertical motion is more difficult than dealing with the horizontal, because the vertical components are highly sensitive to slip distribution, and, in our case, its nonlinear interaction with viscoelastic relaxation. In fact, the views of cutting planes in Figure 6 shows that our model does not perform in the vertical as well as in the horizontal.

This work is almost the first challenge ever to combine laboratory-derived mantle and friction constitutive laws in numerical simulations of post-earthquake deformation at subduction zones. Next challenge should be to improve the fit of the near-field vertical displacements. Now we try to refine the models of the coseismic rupture in another project, which will improve our post-earthquake deformation model in the future.

At the conclusion part L148, we emphasized that improving the vertical component is important part of the possible future work as the following:

“The discrepancy between the simulation and the data, particularly in vertical motions and in some seafloor stations, should be reduced, in principle, by refined models of the coseismic rupture and the in-situ conditions such as initial stress, temperature and confining pressure, properties that are usually constrained for long time scales”

>5. Is there a reason you do not show time series data and fits? I suspect fit to time series is very illuminating about how well this model captures the deformation.

In the original manuscript, we did not consider transient-nonlinear creep because its behavior is still not fully understood. We did not show the time series because the early post-earthquake stage needs discussion on transient creep, while simulating the 2.8-year cumulative post-earthquake deformation does not need to incorporate transient creep, which

is justified by Figure 8 in Sobolev and Muldashev (2017), which implies that introduction of transient creep to deformation calculation using steady-state creep does not make so much difference after several months.

However, as you and the other reviewer gave comments regarding the time series, we redid the simulation incorporating nonlinear-transient rheology with the constitutive relation and the parameters based on Masuti et al. (2016), and present the time series of displacement in a few observation stations (Figure 7). Discussion on the time series is added at L102 as the following:

“Temporal increase of effective viscosity takes place in the relaxation process of coseismic stress (Fig. Fig. eviscolater), which explains well the time series of horizontal displacement in the station MYGI and some land stations that are aligned in the trench normal direction from the epicenter (Figure Fig. timehistory).

The misfit in the station MYGW is likely due to the dominance of the elastic response due afterslip there, which we discuss below.”

Reference:

Sobolev, S. V. & Muldashev, I. A. Modeling Seismic Cycles of Great Megathrust Earthquakes Across the Scales With Focus at Postseismic Phase. *Geochemistry, Geophysics, Geosystems* 18 (2017). URL <http://doi.wiley.com/10.1002/2017GC007230>.

Masuti, S., Barbot, S. D., Karato, S., Feng, L. & Banerjee, P. Upper-mantle water stratification inferred from observations of the 2012 Indian Ocean earthquake. *Nature* 538, 373-377 (2016).

REVIEWERS' COMMENTS:

Reviewer #1 (Remarks to the Author):

The work has been significantly improved by including the transient rheology and a comparison between model-predicted with observed GNSS series. These have addressed my main concerns. I think the paper can be accepted after some minor (mainly wording) revision. I will only give some examples, but the authors should work through the text to make further improvements. In particular, in some places, critiques of previous works are not objective and are overly negative.

Lines 6-9. "Assuming that the flow of mantle rocks is Newtonian, the low viscosity required to explain surface deformation was attributed to a weak lithosphere-asthenosphere boundary [4, 6, 7]". This is not an accurate statement. A weak lithosphere-asthenosphere boundary is only one aspect of these previous models. In reference [4], a model without the weak layer also predicts the first-order pattern of surface deformation. The clause "these findings are at odds with well-established results from mineral physics" is disrespectful and sounds like previous authors made some mistake. For understanding fundamental processes, simplifications are often necessary. The difference between simple and complex models is usually not a matter of right or wrong, but only a matter of research focus. I suggest that the authors rewrite the sentence to make it more accurate and properly recognize the fundamental contribution to our understanding of postseismic deformation made by other researchers.

"Partial melts" has been added to line 52, but the ensuing sentence still talks only about the water hypothesis. The authors should be more careful in maintaining the flow of text. Also, the word "invalidate" is inaccurate and offensive in the present context and should be changed to something like "have not offered direct support for ..."

Maxwell and Kelvin elements suddenly appear in the text. Again, the authors should be more careful in maintaining the flow of text. To the minimum, you should let the reader know that details will be given in the Methods section.

Regarding separating contributions from viscoelastic relaxation and afterslip (Figs. 4 and 6), I still do not understand how it can be done in a nonlinear model. Even if you are only showing the elastic response, it is a response after 2.8 years of nonlinear coupling of the two processes. In the three steps described in section 0.5 under Methods, step 2 is a negotiable approximation, but step 3 is incorrect. However, I am willing to see the paper published without changing this, because it is a presentation issue not affecting the modeling itself (e.g., the left half of Fig. 6 is not affected), and readers can use their own judgement.

Regarding the necessity of 900 m topographic resolution, I am not sure whether the author genuinely does not understand my comment or is purposely trying to evade the subject. The lecture on "verification and validation" in the rebuttal letter is completely irrelevant and disingenuous. Nevertheless, the extra-fine topography is just an overkill and does not affect the model results. It is not worth further discussion.

Reviewer #2 (Remarks to the Author):

This is a significantly improved manuscript. Two major concerns I had were: (1) ignoring fit to time series, and (2) ignoring fit to the vertical. The other reviewer was particularly concerned about neglecting transient viscosity. The revised version of the manuscripts addresses the other reviewer's concern about transient viscosity and addresses my concern (1) about fit to the time series.

The authors have done new simulations including transient viscosity. They show a comparison of fit to time series with and without transient creep in their rebuttal letter. The differences are fairly subtle, but it does appear that the transient viscosity model fits the time series better in the first year.

The addition of Figure 7 (fit to time series) is a big improvement, in my opinion. The addition of transient viscosity is a nice improvement as well, although it doesn't appear to have been a major problem with the previous model.

My overall assessment of this manuscript is similar to the previous version. I think what the authors have done here is impressive and it is a significant improvement over previously published postseismic models for this earthquake. The authors have put together a relatively complete model that includes rate-strengthening afterslip coupled to nonlinear mantle flow. Previous models have taken shortcuts (linear rheology or kinematically imposed afterslip). They demonstrate a significant result – that the previously inferred low viscosity LAB boundary shown in several published models with linear viscosity actually evolves naturally in this more complete model because of the nonlinear dependence of viscosity on stress.

I have read through the entire revision and it looks very clean – I did not see any problems.

Reviewer #1 (Remarks to the Author):

Reviewer #1: The work has been significantly improved by including the transient rheology and a comparison between model-predicted with observed GNSS series. These have addressed my main concerns. I think the paper can be accepted after some minor (mainly wording) revision. I will only give some examples, but the authors should work through the text to make further improvements. In particular, in some places, critiques of previous works are not objective and are overly negative.

Reply: Thank you very much for your positive outlook. We are grateful for your further comments. It is pity that we made you doubt of our genuineness. We made further improvements to the text to better reflect on previous work. Some parts were modified according to the journal's formatting rules.

Reviewer #1: Lines 6-9. "Assuming that the flow of mantle rocks is Newtonian, the low viscosity required to explain surface deformation was attributed to a weak lithosphere-aesthenosphere boundary [4, 6, 7]". This is not an accurate statement. A weak lithosphere-aesthenosphere boundary is only one aspect of these previous models. In reference [4], a model without the weak layer also predicts the first-order pattern of surface deformation. The clause "these findings are at odds with well-established results from mineral physics" is disrespectful and sounds like previous authors made some mistake. For understanding fundamental processes, simplifications are often necessary. The difference between simple and complex models is usually not a matter of right or wrong, but only a matter of research focus. I suggest that the authors rewrite the sentence to make it more accurate and properly recognize the fundamental contribution to our understanding of postseismic deformation

made by other researchers.

Reply: We agree with the reviewer and apologize for the poor choice of word. We consider highly the previous studies on the Tohoku earthquake, which were a source of inspiration for our current work. We modified the abstract and the main text accordingly.

Reviewer #1: "Partial melts" has been added to line 52, but the ensuing sentence still talks only about the water hypothesis. The authors should be more careful in maintaining the flow of text. Also, the word "invalidate" is inaccurate and offensive in the present context and should be changed to something like "have not offered direct support for ..."

Reply: Very good suggestion. We replaced

"recent experiments invalidate high water concentration to explain the low-velocity layer at the lithosphere-asthenosphere boundary."

with

"these findings require explanations consistent with mineral physics".

Reviewer #1: Maxwell and Kelvin elements suddenly appear in the text. Again, the authors should be more careful in maintaining the flow of text. To the minimum, you should let the reader know that details will be given in the Methods section.

Reply: Thank you for the suggestion. We modified so that it does not give impression that Maxwell and Kelvin elements suddenly appear.

Reviewer #1: Regarding separating contributions from viscoelastic relaxation and afterslip (Figs. 4 and 6), I still do not understand how it can be done in a nonlinear model. Even if you are only showing the elastic response, it is a response after 2.8 years of nonlinear coupling of the two processes. In the three steps described in section 0.5 under Methods, step 2 is a negotiable approximation, but step 3 is incorrect. However, I am willing to see the paper published without changing this, because it is a presentation issue not affecting the modeling itself (e.g., the left half of Fig. 6 is not affected), and readers can use their own judgement.

Reply: The approach is correct. We added a subheading “Decomposition of source mechanisms and induced deformation” in the main text and improved the explanation in the subheading “Viscoelastic and afterslip contributions” in Methods further. Please see the newly added sentences in Methods for the definitions of the characters used in the following. The key point is that viscoelastic deformation is linear functional of viscoelastic strain change $\Delta\epsilon_v$. Then it should be clear why we can decompose the post-earthquake displacement as

$$\mathbf{u}_{\text{original}} = \mathbf{G}_d\Delta\mathbf{d} + \mathbf{G}_\epsilon\Delta\epsilon_v$$

Nonlinear coupling, which the reviewer pointed out, takes place in contribution to $\Delta\mathbf{d}$ and $\Delta\epsilon_v$ from the stress imposed by coseismic slip, afterlip and viscoelastic relaxation. These three contributions are nonlinearly coupled and cannot be decomposed, but resulting displacements from $\Delta\mathbf{d}$ and $\Delta\epsilon_v$ can be added linearly.

Reviewer #1: Regarding the necessity of 900 m topographic resolution, I am not sure whether the author genuinely does not understand my comment or is purposely trying to evade the subject. The lecture on “verification and validation” in the rebuttal letter is completely irrelevant and disingenuous. Nevertheless, the extra-fine topography is just an overkill and does not affect the model results. It is not worth further discussion.

Reply: We value the reviewer’s comments and we have tried our best to address all comments thoroughly. It is true that including a realistic topography is not central to the model, but it does not impact the results negatively either. We refine the mesh near the surface to obtain numerical convergence and to improve the accuracy of near-surface synthetic data.

Reviewer #2 (Remarks to the Author):

Reviewer #2: This is a significantly improved manuscript. Two major concerns I had were: (1) ignoring fit to time series, and (2) ignoring fit to the vertical. The other reviewer was particularly concerned about neglecting transient viscosity. The revised version of the manuscripts addresses the other reviewer’s concern about transient viscosity and addresses

my concern (1) about fit to the time series.

Reply: We are pleased that our revision addresses the major concerns from Reviewer #2.

The authors have done new simulations including transient viscosity. They show a comparison of fit to time series with and without transient creep in their rebuttal letter. The differences are fairly subtle, but it does appear that the transient viscosity model fits the time series better in the first year.

Reply: Correct. The addition of transient creep improved the model.

The addition of Figure 7 (fit to time series) is a big improvement, in my opinion. The addition of transient viscosity is a nice improvement as well, although it doesn't appear to have been a major problem with the previous model.

Reply: Transient creep makes a more drastic difference when a linear rheology is used in the first place. As we use a nonlinear rheology, the fit to the data was already quite satisfactory. That said, including transient creep led to a measurable improvement.

My overall assessment of this manuscript is similar to the previous version. I think what the authors have done here is impressive and it is a significant improvement over previously published postseismic models for this earthquake. The authors have put together a relatively complete model that includes rate-strengthening afterslip coupled to nonlinear mantle flow. Previous models have taken shortcuts (linear rheology or kinematically imposed afterslip). They demonstrate a significant result – that the previously inferred low viscosity LAB boundary shown in several published models with linear viscosity actually evolves naturally in this more complete model because of the nonlinear dependence of viscosity on stress.

Reply: This is indeed one key finding of this work.

I have read through the entire revision and it looks very clean – I did not see any problems.

Reply: We thank you for your positive concluding remarks. Your comments have significantly improved the manuscript and led to a more comprehensive model.